environmental chemistry

sodium *p*-perfluorous nonenoxybenzene sulfonate, activated carbons, adsorption, competitive adsorption, regeneration, mechanism

**Author for correspondence:**
Huilan Shi
e-mail: hlshi197701@126.com

This article has been edited by the Royal Society of Chemistry, including the commissioning, peer review process and editorial aspects up to the point of acceptance.

# Adsorption behaviour and mechanism of the PFOS substitute OBS (sodium *p*-perfluorous nonenoxybenzene sulfonate) on activated carbon

Wei Wang[1], Xin Mi[1,2], Huilan Shi[1,2], Xue Zhang[1,2], Ziming Zhou[1,2], Chunli Li[1] and Donghai Zhu[1]

[1]State Key Laboratory of Plateau Ecology and Agriculture, and [2]Eco-environmental engineering college, Qinghai University, Xi'ning, Qinghai Province 810016, People's Republic of China

(iD) HS, 0000-0003-0801-7520

Perfluorooctane sulfonate (PFOS) was listed as a persistent organic pollutant by the Stockholm Convention. As a typical alternative to PFOS, sodium *p*-perfluorous nonenoxybenzene sulfonate (OBS) has recently been detected in the aquatic environment which has caused great concern. For the first time, the adsorption behaviour and mechanism of OBS on activated carbon (AC) with different physical and chemical properties were investigated. Decreasing the particle size of AC can accelerate its adsorption for OBS, while AC with too small particle size was not conducive to its adsorption capacity due to the destruction of its pore structure during the mechanical crushing process. Intra-particle diffusion had a lesser effect on the adsorption rate of AC with smaller particle size, higher hydrophilicity and larger pore size. Reactivation of AC by KOH can greatly enlarge their pore size and surface area, greatly increasing their adsorption capacities. The adsorption capacity of two kinds of R-GAC exceeded 0.35 mmol g$^{-1}$, significantly higher than that of other ACs. However, increasing the hydrophilicity of AC would decrease their adsorption capacities. Further investigation indicated that a larger pore size and smaller particle size can greatly enhance the adsorptive removal of OBS on AC in systems with other coexisting PFASs and organic matter due to the reduction of the pore-blocking effect. The spent AC can be successfully regenerated by methanol, and it can be partly regenerated by hot water and NaOH solution. The percentage of regeneration for the spent AC was 70.4% with 90°C water

temperature and up to 95% when 5% NaOH was added into the regeneration solution. These findings are very important for developing efficient adsorbents for the removal of these newly emerging PFASs from wastewater and understanding their interfacial behaviour.

## 1. Introduction

Per- and polyfluoroalkyl substances (PFASs) in aquatic environments have attracted a great deal of attention due to their high environmental persistence, worldwide prevalence and widespread use for over 60 years in various commercial and industrial applications [1,2]. Perfluorooctane sulfonate (PFOS) is the most commonly studied of the PFASs, and has been found worldwide in the aquatic environment, and even in human blood [3,4]. PFOS was considered to be a threat to the natural environment due to its high toxicity, bioaccumulation and stability, and it was listed as a persistent organic pollutant in May 2009 [2,5]. Fortunately, the decrease of PFOS concentration in human blood observed in developed countries suggested that those restriction actions have reduced the human exposure risks [6–8], and China is already beginning to control it. However, due to their unique properties, it is difficult to replace perfluorinated compounds with fluorine-free compounds. Some short-chain alternatives with a similar organo-fluorinated structure to PFOS are considered safer than the longer chain PFOS, and have been mass produced as alternative substances. They still, however, show high stability under environmental conditions and are toxic to the aquatic environment [9–11]. Moreover, these novel PFASs have recently been found worldwide in the environment and also have an influence on human bodies [12]. Even so, there is little information available about their environmental behaviour at the solid–liquid interface.

Sodium $p$-perfluorous nonenoxybenzene sulfonate (OBS), with a benzene ring moiety, belongs to the PFASs group. OBS has been used in various commercial and industrial applications, e.g. in the production of high-efficiency film-forming fluoroprotein foams, as oil exploitation agents, in steel plate cleaning, photographic film and printing (http://www.kalfchina.com/product/showproduct.php?lang=cn&id=36). Due to the restriction of the use of PFOS in consumer products and industrial processes, OBS is becoming a popular additive with an estimated total production volume of about 3500 t per year in China [13]. Recently, its occurrence and environmental behaviour has become a hot topic. It was found that the maximum concentration of OBS in surface water was $32\,000\,ng\,l^{-1}$ detected near an old oilfield [8]. OBS was also found to be not as safe as expected. The normal function of soil microorganisms was greatly affected by OBS pollution [14]. OBS was proved to have similar acute toxicity when compared with PFOS according to the Globally Harmonized System criteria, and caused the mortality of fish and tadpoles with $LC_{50}(96\,h)$ values of $25.5\,mg\,l^{-1}$ and $28.4\,mg\,l^{-1}$, respectively [8,15]. Because of a serious concern regarding its further spread in aqueous environment and consequent risk to biological and human health, it is very important to study the environmental transport, environmental fate and removal of OBS from water and wastewater using effective techniques. To the best of our knowledge, there are only two papers discussing OBS removal from water using an oxidation method and aeration-foam collection [13,16]. OBS can be decomposed under UV and UV/$H_2O_2$, but will produce complex by-products raising concern about its potential toxicity [13]. Aeration-foam collection is effective for OBS removal from water to the $\mu g\,l^{-1}$ level; however, further treatment is also needed [16]. Wastewater treatment plants are unable to remove OBS from wastewater [17]. Therefore, it is necessary to develop effective methods to remove OBS from water.

It has recently been proved that adsorption is an effective and economical technology for PFOS removal from water [18], and activated carbon (AC) was considered as one of the most effective adsorbents for the removal of PFOS from water [5,19,20]. There is, however, a lack of information regarding OBS adsorption on ACs. Compared with PFOS, the OBS structure contains one ether unit and aromatic moiety among the C–F chain, this gives OBS different properties and possibly a corresponding difference in its adsorption behaviour and on-surface adsorption mechanism. OBS may be adsorbed on the surface of AC via not only hydrophobic interaction (the main interaction of PFOS adsorbed on AC) but also π–π interaction. Normally, different PFASs coexist in the polluted water, but the removal of target PFASs in the system has not been fully studied. Previous studies indicated that the OBS-contaminated environment contained high concentrations of PFOS, PFOA and other short-chain PFASs [8,21]. Therefore, it is necessary to know whether ACs are effective for the removal of OBS from water or PFASs that also coexist in the water.

Our study is the first one to investigate the adsorption behaviour and mechanism of OBS on AC. The sorption kinetics, isotherms, and effect of competing PFASs, solution pH and ionic strength were investigated. Seven ACs with different properties were prepared, and used to compare their different adsorption behaviours for OBS. Finally, different regenerated solutions were developed to regenerate the spent AC.

# 2. Material and methods

## 2.1. Chemicals and materials

OBS was purchased from Wengjiang reagent Co. Ltd (Guangzhou, China), and its structural formula is shown in electronic supplementary material, table S1. Perfluorobutane sulfonate (PFBS, potassium salts), perfluorobutanoic acid (PFBA), perfluorooctanesulfonate (PFOS, potassium salt) and perfluorooctanoate (PFOA, sodium salt) were obtained from Sigma-Aldrich Co. Shell-activated carbon was obtained from Jingke Activated Carbon Co. (Tangshan, China). The ultrapure water was produced by a Milli-Q integral water purification system (Millipore, USA). The other chemicals were all analytical reagent grade.

## 2.2. Preparation of activated carbons

To fully consider the influence of different forms of ACs with various physical–chemical properties on the adsorption process of OBS, seven types of ACs including granular-activated carbon (GAC), powdered-activated carbon (PAC), ultrafine AC (UAC), two oxidized GAC (O1-GAC and O2-GAC) and two reactivated GAC (R1-GAC and R2-GAC) were selected as target adsorbents to study their adsorption behaviours. GAC, PAC, UAC, O1-GAC, O2-GAC, R1-GAC and G2-GAC were all prepared from the same shell-activated carbon. PAC was made by mechanical milling and sieving with particle size of 20–25 mesh. The UAC was prepared by ball milling in a planetary ball mill (Nanjing University Instrument Co., China) with stainless steel vials (80 ml) and balls (diameter = 5.60 mm) for 3 h. To obtain GAC with enlarged pore size, GAC was heated to 900°C under $N_2$ in a tubular furnace. Before heating, the GAC was impregnated by KOH solution at KOH/C mass ratios of 1 (R1-GAC) or 4 (R2-GAC) for 48 h, following by drying at 100°C for 24 h in an air oven and then heated at 900°C under $N_2$ for 1.5 h. Finally, the reactivated GAC particles were washed by 1 mol l$^{-1}$ HCl solution and Milli-Q water until the pH of the washing water was approximately 7. To prepare GAC with a highly oxidized surface, 1 g GAC was added into a 20 ml reaction vessel with 15 ml of 23% (O1-GAC) or 70% w/w $HNO_3$ solution (O2-GAC), and the oxidation treatment was conducted for 4 h. Following this, the reaction by-products and residual oxidants were removed by repeatedly washing with milli-Q water until the suspension reached approximately neutral pH.

## 2.3. Characterization of adsorbents

The specific surface area (SSA) and pore size distribution of the seven ACs were characterized by $N_2$ adsorption at 77 K in a gas adsorption instrument (Autosorb iQ, Quantachrome Corp., USA). The surface morphologies of the virgin ACs, oxidized ACs and reactivated ACs were observed by a scanning electron microscopy (SEM, JSM-6460LV, JEOL, Japan). The zeta potentials of the ACs were determined by a zeta potential instrument (Delsa Nano C, Beckman Coulter, USA). The particle size distribution of UAC was measured using laser particle size distribution (Dandong Baxter Instruments Co., China). The elemental composition of the prepared ACs was measured using an elemental analyser (EA3000, Italy).

## 2.4. Adsorption experiments

Prior to the adsorption experiment, the OBS solution was mixed for 20 min using ultrasound. The adsorption experiments were carried out in an orbital shaker at a rotating speed of 150 r.p.m. and a temperature of 25°C. The adsorption kinetic experiments were conducted in 180 ml of OBS solution with an initial concentration of 20 mg l$^{-1}$ containing 15 mg of the different ACs. The adsorption isotherms were continuously agitated in 80 ml OBS solution containing different initial concentrations (10–250 mg l$^{-1}$). Experiments examining the effect of ionic strength and pH were carried out in 80 ml of 20 mg l$^{-1}$ OBS solution. NaCl or $CaCl_2$ (0–100 mmol l$^{-1}$) was added into the adsorption system to investigate the effect of ionic strength, and solution pH values were adjusted to 2–8 (without pH

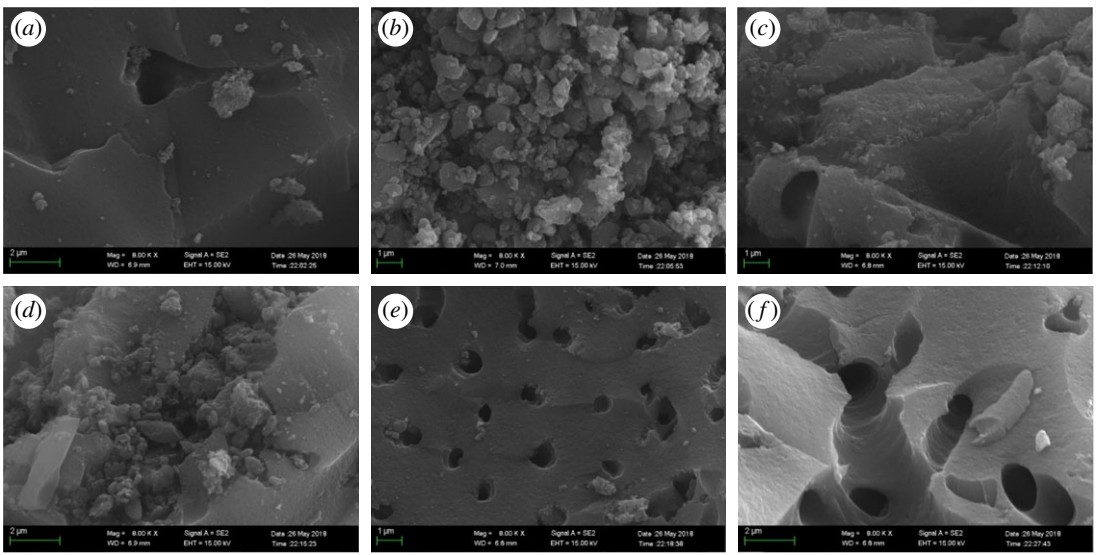

**Figure 1.** SEM images of virgin AC (*a*), UAC (*b*), O1-GAC (*c*), O2-GAC (*d*), R1-GAC (*e*) and R2-GAC (*f*) at the magnitude of ×8000.

adjustment during the adsorption process) to investigate the influence of pH on the adsorption process. In the investigation of coexisting PFASs on the adsorption of OBS on the ACs, the competition adsorption experiments were conducted with the same initial concentration of 0.032 mmol $l^{-1}$ (PFOS, PFOA, PFBS, PFBA and OBS). To study the application of ACs in real wastewater, municipal wastewater was collected from the Xining wastewater treatment plant in Xining, Qinghai, People's Republic of China (pH = 8.01; TOC = 121.7 mg $l^{-1}$). Suspended matter was removed using a 0.45 µm membrane, and OBS adsorbate was added to obtain a concentration of 20 mg $l^{-1}$. All experiments were conducted in duplicate.

## 2.5. Regeneration experiments

After the adsorption of 10 mg GAC in 80 ml OBS solution, the spent GAC was taken out and regenerated in 50 ml of regenerating solution. The regenerating studies were conducted in an oil bath shaker with a rotating speed of 150 r.p.m. for 12 h, and the regeneration temperature was varied from 25°C to 90°C.

## 2.6. Analytical methods

Samples were centrifuged at 13 000 r.p.m. for 10 min and the supernatant removed for analysis. OBS concentration was determined by high-performance liquid chromatography (Agilent 1260 Infinity II, Agilent Technologies, USA) equipped with UV/Vis detector operating at 220 nm and 275 nm and a TC-C18 column (4.6 × 250 mm i.d., particle size of 5 µm; Agilent Technologies, USA). A methanol/0.02 M dihydrogen phosphate buffer solution (8 : 2, v/v) was used as the mobile phase at a flow rate of 1.0 ml $min^{-1}$. The mobile phase was mixed using ultrasound for 30 min to fully blend and remove bubbles. Deionized (DI) water was used for blank determination. The LOD of HPLC-UV was 0.20 mg $l^{-1}$, and the analysis time of each sample was 8.5 min.

# 3. Results and discussion

## 3.1. Characterization of prepared activated carbon

SEM was used to observe the surface morphologies of the different ACs, shown in figure 1. Electronic supplementary material, figure S1 and figure 1*b* show that the major particle diameter of the UAC is approximately 1 µm. Compared with GAC, the surface of O1-GAC and O2-GAC are rougher and fluffier, especially as O2-GAC, with a stronger degree of oxidation, has some micrometre-sized aggregates attached to the surface (figure 1*a,c,d*). After activation using a KOH/C ratio of 1, some pores of less than 1 µm were formed in the surface of the R1-GAC (figure 1*e*). When the reactivation conditions were changed to a KOH/C ratio of 4, most of the pores on the AC surface were enlarged to 1–2 µm (figure 1*f*). In general, R-GAC also has a heterogeneous pore structure similar to AC, but

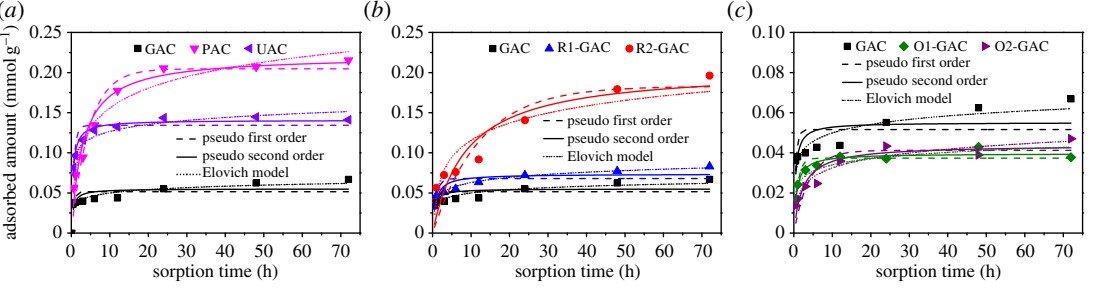

**Figure 2.** Adsorption kinetics of OBS on different activated carbons as well as modelling using the pseudo-first-order, the pseudo-second-order, Elovich kinetic models.

**Table 1.** Characteristics of the activated carbons used in this study.

| adsorbent | surface area (m² g⁻¹) $S_{BET}$ | pore volume cc/g | elemental composition (%) C | H | N | O | (O + N)/C (%) | particle size (mesh) |
|---|---|---|---|---|---|---|---|---|
| GAC | 670.7 | 0.36 | 86.45 | 1.82 | 0.19 | 4.62 | 5.57 | 20–25 |
| PAC | 765.1 | 0.51 | 86.45 | 1.82 | 0.19 | 4.62 | 5.57 | 80–150 |
| UAC | 632.5 | 0.43 | 85.74 | 2.01 | 0.17 | 7.79 | 9.29 | electronic supplementary material, figure S1 |
| R1-GAC | 1108.6 | 0.62 | 91.63 | 1.19 | 0.13 | 3.45 | 3.91 | 20–25 |
| R2-GAC | 1705.1 | 0.89 | 88.41 | 1.35 | 0.03 | 5.42 | 6.17 | 20–25 |
| O1-GAC | 727.9 | 0.44 | 81.64 | 2.35 | 0.78 | 7.95 | 10.70 | 20–25 |
| O2-GAC | 725.5 | 0.44 | 78.78 | 2.32 | 0.90 | 10.81 | 14.87 | 20–25 |

the pore size and the number of pores of R-GAC were increased after activation. The enlarged pore sizes are favourable for the diffusion of molecular OBS into the granular adsorbent.

The pore size distributions and Brunauer–Emmett–Teller (BET) surface areas of the seven ACs are shown in electronic supplementary material, figure S2 and table 1. The SSA of GAC is a little lower than that of PAC but higher than UAC (table 1). The SSA of UAC is lower, because its pore structure has been destroyed during such high strength mechanical crushing process [22]. Due to the large particle size of GAC, some internal pores which can adsorb $N_2$ are unavailable and the low-intensity mechanical disruption facilitates the exposure of these pores, resulting in a larger SSA. However, too intensive mechanical crushing will destroy the pore structure of the ACs and reduce their SSA. When the ACs were reactivated at different KOH/C ratios, their SSA increased steeply from 670.7 to 1705.1 m² g⁻¹ with increasing the KOH/C ratio to 4 (table 1). As shown in electronic supplementary material, figure S2(b), the number of pores of reactivated GAC with sizes less than 4 nm greatly increased. The molecular length of OBS is 1.26 nm, therefore, the enlarged pores in the range of 2–5 nm would be more conducive to the removal of OBS from water. It is notable that the SSA and pore size distribution of GAC changed little after the oxidation treatment (electronic supplementary material, figure S2(c) and table 1). Electronic supplementary material, figure S3 shows the FT-IR spectra of R1-GAC and R2-GAC. There are hydroxyl and carboxyl oxygen-containing functional groups on the surface of R-GAC. The detailed properties of seven ACs are summarized in table 1.

## 3.2. Adsorption kinetics

Figure 2 shows the adsorption kinetics of OBS on different ACs. UAC exhibited highest adsorption during the initial 5 h, followed by PAC and GAC (figure 2a). Adsorption equilibrium was achieved after 24 h for PAC and UAC, while at least 72 h was needed for GAC. The longer equilibration time for GAC was similar to our previously reported data for the adsorption of organophosphate flame retardants on AC [23]. Due to the big size of GAC and the big molecular size of OBS, it took a long

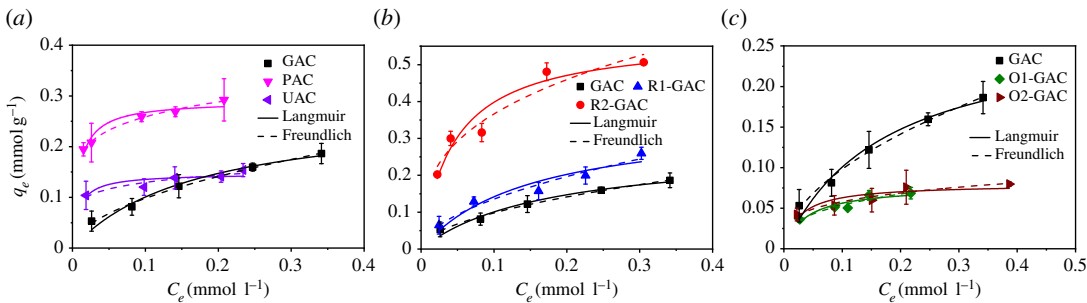

**Figure 3.** Adsorption isotherms of OBS on different activated carbons as well as modelling using the Langmuir and Freundlich isotherm models.

time for OBS to diffuse into the long pores of GAC. ACs with a larger particle size have deeper pores and less exposed sites for OBS adsorption, resulting in a correspondingly slower adsorption process. The kinetic curve of R1-GAC and R2-GAC exhibits similar trends to GAC (figure 2b), and the adsorption process proceeds slowly between 20 and 72 h due to the slow diffusion of the adsorbate molecules into the long pores of the granular ACs. Figure 2c shows the adsorption kinetics of AC with different degrees of oxidation. The required contact time for O1-GAC and O2-GAC to reach adsorption equilibrium was less than 48 h, which is faster than that of GAC. The quicker adsorption equilibrium of ACs possessing a more hydrophilic skeleton could be attributed to the fact that the diffusion of a hydrophobic organic pollutant into the adsorption pores is easier due to the faster transfer of water molecules in the hydrophilic surface [24].

To further compare the adsorption kinetics of OBS on the seven ACs, five kinetic models: pseudo-first-order, pseudo-second-order, Elovich, intra-particle diffusion and Boyd model were used to fit the data. These are described in the electronic supplementary material. The results of fitting the experimental data to the models are shown in the figure 2 and electronic supplementary material, figure S4, figure S5 and table S2. Based on the correlation coefficient ($R^2$), the pseudo-second-order model fitted the experimental data for OBS adsorbed on the seven ACs better than the pseudo-first-order model, suggesting that the chemical adsorption between ACs and OBS was possibly produced in the adsorption process. Moreover, UAC exhibited a much higher initial adsorption rate ($v_0$) than the other ACs, highlighting the advantage of ultrafine AC for quick removal of organic pollutant from water. The Elovich model is commonly used for clarifying if there is active chemisorption, and this model with higher $R^2$ value compared with the other kinetic models further indicates chemical adsorption between ACs and OBS. Normally, the adsorption mechanism depends not only on the physical–chemical properties of the adsorbent but also the transport process of adsorbate [25]. The results of the intra-particle diffusion model used to describe the experimental data are shown in electronic supplementary material, figure S4 and table S2. The adsorption curves can be divided into two stages, and the fitted results of the kinetic parameters of first stage are given in electronic supplementary material, table S2. The $c_{n1}$ of the fitted results followed the order of $c_{n1}(GAC) < c_{n1}(PAC) < c_{n1}(UAC)$, $c_{n1}(GAC) < (O1\text{-}GAC) < (O2\text{-}GAC)$ and $c_{n1}(GAC) < c_{n1}(R1\text{-}GAC) < c_{n1}(R2\text{-}GAC)$, indicating the intra-particle diffusion had lesser effect on the adsorption rate of AC with smaller particle size, higher hydrophilicity and larger pore size. The surprisingly small $c_{ni}$ value of GAC and large $c_{ni}$ value of UAC suggested that intra-particle diffusion is the main rate-limiting step for OBS absorption on GAC and boundary diffusion is the main rate-limiting factor for OBS absorption on UAC. The Boyd model has been widely used for studying the mechanism of adsorption [26], and electronic supplementary material, figure S5 showed the Boyd plots for the first 24 h adsorption of OBS on ACs. The plots did not pass through the origin or show a linear segment before sorption equilibrium, suggesting that the rate of adsorption was not only controlled by pore diffusion in the initial period and chemical reaction also controlled the rate of adsorption.

## 3.3. Adsorption isotherms

The adsorption isotherms for OBS on the different ACs are shown in figure 3, together with the Langmuir and Freundlich models used to describe the isotherm data [27,28]. As shown in table 2 and figure 3, the adsorption capacity for OBS on GAC was lower than PAC but higher than UAC, while UAC exhibited a higher level of adsorption at the equilibrium concentration below 0.15 mmol l$^{-1}$. These differences can be

**Table 2.** Calculated parameters of the Langmuir and Freundlich equations for OBS adsorption on seven ACs.

| adsorbent | Langmuir parameters[a] | | | Freundlich parameters[b] | | |
|---|---|---|---|---|---|---|
| | $q_m$ (mmol g$^{-1}$) | $b$ (l mmol$^{-1}$) | $R^2$ | $K_f$ (mmol$^{1-1/n}$ l$^{1/n}$ g$^{-1}$) | $n$ | $R^2$ |
| GAC | 0.2738 | 5.3 | 0.959 | 0.329 | 1.91 | 0.991 |
| PAC | 0.2904 | 116.9 | 0.912 | 0.370 | 6.45 | 0.993 |
| UAC | 0.1470 | 116.8 | 0.668 | 0.184 | 6.58 | 0.868 |
| R1-GAC | 0.3539 | 6.7 | 0.889 | 0.467 | 1.86 | 0.947 |
| R2-GAC | 0.5764 | 22.3 | 0.903 | 0.778 | 3.06 | 0.912 |
| O1-GAC | 0.0773 | 26.7 | 0.806 | 0.114 | 3.06 | 0.897 |
| O2-GAC | 0.0796 | 37.8 | 0.718 | 0.102 | 4.12 | 0.906 |

[a]Langmuir model: $q_e = q_m C_e/(1/b + C_e)$.
[b]Freundlich model: $q_e = K_f C_e^{1/n}$.

partially attributed to the different surface area and pore distributions of the different ACs, shown in table 1. The slightly higher surface area of PAC cannot explain its almost double sorption capacity compared with that of GAC (figure 3a). The higher number of available sorption sites on PAC may cause the differences due to the more obvious pore-blocking effect on OBS adsorption on GAC. At low OBS equilibrium concentrations, there were more available sorption sites than molecular OBS, and UAC showed higher sorption amounts than GAC due to the shorter pores and more easily assessable sorption sites in UAC. At high OBS equilibrium concentrations, the lower surface area and higher degree of oxidation of UAC (table 1) would cause its limited sorption capacity. Indeed, it can be seen from figure 3c that O1-GAC and O2-GAC with higher hydrophilicity (based on (N + O)/C value) showed much lower adsorption capacities than that of GAC. However, the adsorption capacities changed little with any further increase in the degree of oxidation. A previous study proved that hydrophobic interaction is the key factor that causes PFASs to be adsorbed on carbon materials, increasing the hydrophilic surface area of adsorbents will therefore, significantly decrease their adsorption capacity [29]. The unchanged adsorption capacities of O2-GAC with any further increase in its degree of oxidation was possibly due to hydrogen bonding overcoming the further decrease of hydrophobic interaction [18]. As can be seen from figure 3b, a higher KOH/C ratio during the reactivated process will increase adsorption capacities, and R2-GAC had approximately two times the adsorption capacity of GAC. KOH can react with ACs to produce some pores (figure 1), and higher KOH/C ratios would accelerate the process, which is conducive to OBS adsorption on ACs.

The Langmuir equation assumes that during the adsorption process there is monolayer coverage of the adsorbates on the adsorbents with no interactions between the adsorbate molecules [30,31]. Previous studies reported the adsorption of PFOS on ACs followed the Langmuir model [19,20]; however, when comparing $R^2$ values, the Freundlich model fitted the adsorption of OBS on ACs better in the present study. In contrast to PFOS, OBS possesses an aromatic moiety, and it is this that may generate the intra-molecular interaction between the aromatic compounds during the adsorption process [32]. The $n$ value of the Freundlich model is the nonlinear indicator, and $n$ values in the range of 1.862–6.447 suggest the occurrence of nonlinear adsorption for all the adsorption isotherms in this study. To help clarify the complex adsorption mechanisms, the D–R model was used to fit the isotherm data, shown in electronic supplementary material, figure S6. The value of mean sorption energy ($E$; kJ mol$^{-1}$) provides information about physical and chemical adsorption, which was calculated according to the D–R model [33]. $E$ values less than 8 kJ mol$^{-1}$ suggest physical adsorption; values greater than 8 kJ mol$^{-1}$ suggest chemical adsorption [34]. In this study, the $E$ values were all above 8 kJ mol$^{-1}$ (electronic supplementary material, figure S6), indicating that the adsorption of OBS on ACs involves chemical interaction, which further supports the theory outlined above.

## 3.4. Effect of solution pH and ionic strength

Data on the effect of solution pH on the removal of OBS by AC are shown in figure 4a. The percentage of OBS removed by PAC decreased significantly with increasing solution pH. The zeta potentials of PAC

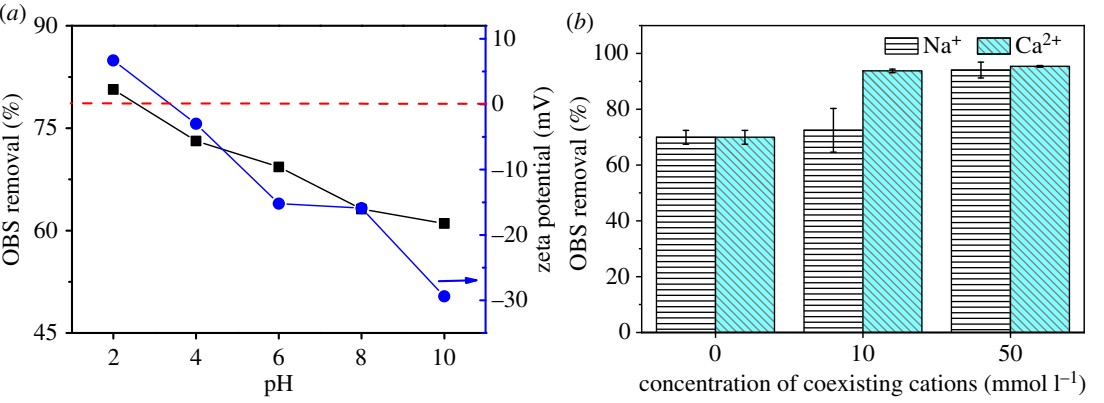

**Figure 4.** Effect of solution pH (a) and cationic concentrations (b) on OBS removal by PAC; (a) also shows the zeta potentials of PAC at different pHs.

surfaces at solution pHs ranging from 2 to 10 are shown in figure 4a, with the zeta potentials decreasing with increasing solution pH in this range. When the solution pH was 2, the PAC surface was positively charged. Since molecular OBS exists as an anion in solution at pH 2, the enhanced adsorption capacities at a solution pH of 2 may be attributed to the electrostatic attraction between the positively charged surface of PAC and OBS. When the solution pH was increased from 4 to 10, the PAC surface became negatively charged. The decrease in OBS removal by PAC was due to the increase of electrostatic repulsion with increasing solution pH.

NaCl and $CaCl_2$ were used to study the effect of ionic strength on the removal of OBS by AC, the results of which are shown in figure 4b. The removal of OBS by PAC increased when increasing the $Na^+$ concentration from 0 to 50 mmol $l^{-1}$, and the higher increase in the removal of OBS in the presence of $Ca^{2+}$ could be attributed to the stronger salting-out effect of $Ca^{2+}$ when compared with $Na^+$ [35]. This is consistent with previous work [36] which showed that $Ca^{2+}$ showed a stronger influence than $Na^+$ on the adsorption of PFASs on PAC. Moreover, the solubility of OBS decreased sharply at high salt concentration, behaviour which is similar to PFOS [37]. The aggregation of PAC can be formed due to the squeezing-out effect derived from ionic strength [38], and electrostatic screening can also cause the weak electrostatic repulsion among PAC particles and aggregation of PAC [39]. These two opposing effects caused by the solutions ionic strength were possibly involved in the OBS adsorption on ACs. However, since the salt-out effect exhibited a more significant effect than PAC aggregation, OBS removal rate increased with the increase of ionic strength.

## 3.5. Adsorbent regeneration and applicability in coexisting systems

ACs containing adsorbed PFOS can be regenerated using hot water, organic solvents, salt solutions and advanced oxidation [5,19,40]. Moreover, the ACs showed lower adsorption capacities for OBS at higher solution pH. Increasing the regenerated solution pH may, therefore, cause the desorption of adsorbed OBS. In this study, the spent AC was regenerated using hot water, NaOH, ethanol and methanol. The temperature of the regenerated solution was found to greatly affect the regeneration efficiency of spent AC (figure 5a). The percentage of regeneration for the spent AC increased from 40.1% to 70.4% with increasing water temperature from 25°C to 90°C. Our previous study showed that the hydrophobic surfactant adsorbed on the air bubbles found on the AC surfaces, and this is primarily responsible for the adsorption of the surfactant on AC [40]. Increasing the water temperature can cause the desorption of these air bubbles and a corresponding desorption of PFOS [40]. Furthermore, increasing the dosage of NaOH can greatly increase the regeneration percentage of the spent AC due to the desorption of OBS at higher solution pH. When the addition of NaOH was increased to 5%, the regeneration percentage was higher than 95%. It was also found that OBS-adsorbed AC was regenerated by organic solvents, and a higher concentration of the organic solvent showed a higher regeneration percentage. Methanol showed a higher regeneration performance than ethanol, while increasing the temperature caused a decrease in the regeneration percentage due to the volatilization of the organic solvent.

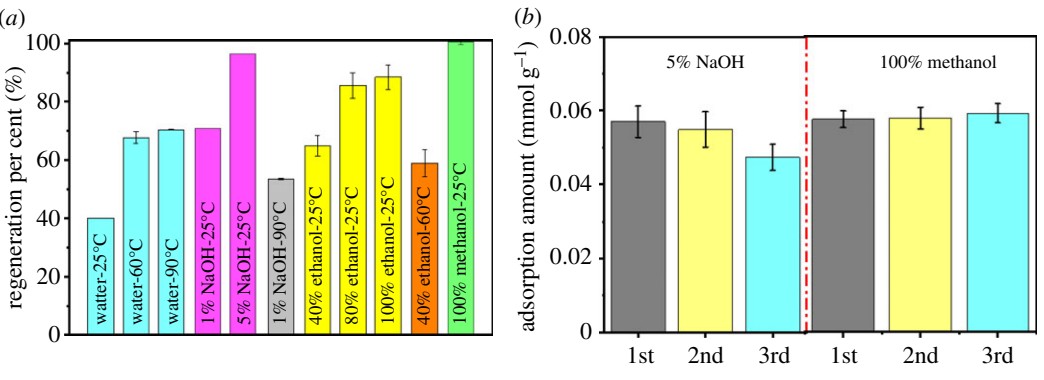

**Figure 5.** Regeneration per cent of the spent GAC under different conditions (*a*) and OBS removal in three successive adsorption cycles (*b*).

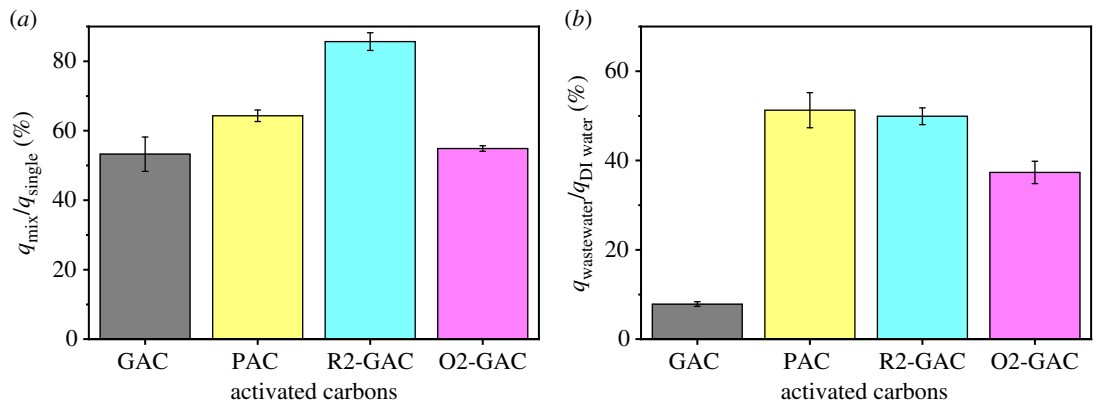

**Figure 6.** Effect of solution coexisting PFASs in pure water (*a*) and coexisting matters in real wastewater (*b*) on OBS removal by different ACs.

Based on the data shown in figure 5*a*, 5% NaOH and methanol were selected as the best regeneration agents, and the regenerated AC was re-used for three successive adsorption-desorption cycles (figure 5*b*). The amounts of OBS absorbed on the regenerated AC were consistent over the three successive regeneration cycles using methanol as the regeneration agent. However, the adsorption amount of OBS regenerated by 5% NaOH decreased in the following adsorption cycles. Evidently, after OBS absorption, the spent AC can easily be regenerated by methanol and shows good reusability for OBS removal from water.

As previously mentioned, water containing OBS often has coexisting PFOS, PFOA and other short-chain PFASs. The ratio ($q_{mix}/q_{single}$) between the adsorption capacity of the ACs in the coexisting PFASs solution (OBS, PFOS, PFOA, PFBS and PFBA) and the single OBS solution was calculated to evaluate the influence of coexisting PFASs on the various ACs. The presence of coexisting PFASs had a great effect on the adsorption of OBS on ACs (figure 6*a*). The adsorption capacity of GAC decreased by approximately 47% in the coexisting system, while PAC decreased by approximately 36%. The pore-blocking effect caused by the long-chain PFOA and PFOS is more obvious with GAC than PAC [20], causing the difference in their adsorption performance in the coexisting system. Furthermore, R-GAC showed the highest $q_{mix}/q_{single}$ value, which showed that increasing the pore size of GAC can greatly enhance its adsorption performance in this system, and confirmed that the intra-particle diffusion had a smaller effect on OBS adsorption with R-GAC than GAC.

To further study the practical application of ACs in wastewater treatment, OBS was added to the influent of municipal wastewater, and the ratio ($q_{wastewater}/q_{DI\ water}$) between the adsorption capacity of the ACs in wastewater and DI water was calculated (figure 6*b*). Increasing the pore size and decreasing the particle size of the ACs will greatly enhance the adsorption performance of ACs in real wastewater. The coexisting substances in the wastewater can compete with OBS for the sorption sites on the ACs and large molecular compounds will block the pores of the ACs, resulting in lower sorption amounts. This blocking effect might have greater effect on GAC adsorption, with its larger particle size and smaller pore size, compared with PAC and R2-GAC, causing its observed negligible adsorptive removal for OBS. Moreover, the GAC showed a lower ($q_{wastewater}/q_{DI\ water}$) value than

O2-GAC, which was consistent with our previous study [24] that hydrophobic organic matter in the wastewater may prefer to interact with the hydrophobic surfaces of adsorbents, resulting in their lower adsorption removal for target compounds.

# 4. Conclusion

In this study, the feasibility of using ACs to remove OBS from aqueous solution was investigated. The adsorption capacity and adsorption rate of OBS on ACs were dependent on their particle size, pore size and hydrophobicity. UAC exhibited the fastest initial adsorption rate and its adsorption equilibrium was reached in 24 h whereas at least 72 h was needed for GAC. Increasing the hydrophilicity of the ACs was found to accelerate their adsorption processes. Adsorption isotherms showed that increasing the pore size and decreasing the particle size of the ACs greatly increased their adsorption capacities, while too small a particle size was unfavourable for OBS adsorption due to the destruction of the pore structure during the mechanical crushing process. Increasing the hydrophilicity of ACs would also decrease their adsorption capacities. R2-GAC had a highest adsorption capacity for OBS with a value of $0.5764 \, \text{mmol} \, \text{g}^{-1}$ according to the Langmuir fitting model. In addition, increasing pore size and deceasing particle size can greatly enhance the removal of OBS from wastewater by ACs. Based on the adsorption results, hydrophobic and electrostatic mechanisms are proposed to be involved in the OBS adsorption process. Additionally, OBS-adsorbed ACs can be successfully regenerated by 100% methanol, and can be re-used three times without loss of their adsorption capacities. Subsequent work is needed to develop novel adsorbents to achieve the selective removal of these emerging PFASs.

Data accessibility. The original database of the work is available in the electronic supplementary material.

Authors' contributions. W.W., X.M., X.Z. and Z.Z. carried out the adsorption experiment; W.W. and H.S. finished the writing; C.L. and D.Z. gave suggestions and modified the manuscript.

Competing interests. We declare we have no competing interests.

Funding. We thank the independent subject of State Key Laboratory of Plateau Ecology and Agriculture (grant no. 2018-ZZ-1), the Project of Qinghai Science & Technology Department (grant nos. 2018-ZJ-Y01, 2019-ZJ-935Q), the International Cooperation Project of Qinghai Province National Science Foundation (grant no. 2017-HZ-810) and the National Natural Science Foundation of China (grant no. 21868030) for financial support.

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
