## [Reviewer comments · Royal Society Open Science]

Review History

RSOS-191069.R0 (Original submission)

Review form: Reviewer 1

Is the manuscript scientifically sound in its present form?

Yes

Are the interpretations and conclusions justified by the results?

Yes

Is the language acceptable?

Yes

Do you have any ethical concerns with this paper?

No

Have you any concerns about statistical analyses in this paper?

Yes

Recommendation?

Accept with minor revision (please list in comments)

Comments to the Author(s)

This article investigated the adsorption behavior and mechanism of sodium p-perfluorooxybenzene sulfonate (OBS) on activated carbons with different physicochemical properties. The removal of OBS from water using activated carbons was attractive to readers. Adsorption kinetics, adsorption isotherms, effects of pH and ionic strength have been examined, and adsorbent regeneration and applicability in co-existing matter systems have also been investigated. The topic is relatively new, which is the first time to examine the adsorptive removal of OBS using activated carbons. This work is important for the practical engineering application of adsorption for removing emerging contaminants. I recommend for publication after minor revision according to the following comments.

1. The abstract could be improved by providing specific information regarding the quantitative data. For example, the adsorption capacity of R-AC? the regeneration percent of hot water and NaOH solution?
2. OBS was a typical alternative to PFOS, and its structural formula should be given, which is very important to know the structural difference compared with PFOS?
3. This paper evaluated the influence of co-existing PFASs on the adsorption of OBS on ACs. OBS in actual water environment always coexists with other PFASs. What is the selectivity of activated carbons for the target OBS?
4. Page 4, line 6, better to use 'Per- and polyfluoroalkyl substances (PFASs)', instead of 'Perfluoroalkyl and polyfluoroalkyl substances'.
5. Page 9, line 21-22, better to give a more detailed description of the analytical methods, e.g., blanks, the LOD of HPLC-UV, the preparation of mobile phase, the sample analysis time, etc.
6. Page 14, line 14, change "previous studies" to " a previous study".
7. Page 15, Lines 28-29 mention two methods previously tested for regeneration of spent AC used to treat PFOS. Better to consider citing additional research related to spent AC that was specifically used to treat PFAS.
8. Fig. 3(a), UAC, not BAC. Fig. 3(c), O1-GAC, not Q1-GAC.
9. The adsorption isotherms of OBS was reported, and the relative adsorption references should be cited, such as *Nanoscale*, 2016, 8, 15978-15987, *Inorganic Chemistry* 2019, 58, 11, 7255-7266, *Mater. Chem. Front.*, 2019,3, 224-232. *Dalton Trans.*, 2018,47, 2791-2798,

Review form: Reviewer 2

Is the manuscript scientifically sound in its present form?

Yes

Are the interpretations and conclusions justified by the results?

Yes

Is the language acceptable?

Yes

Do you have any ethical concerns with this paper?

No

Have you any concerns about statistical analyses in this paper?

Yes

Recommendation?

Accept with minor revision (please list in comments)

Comments to the Author(s)

The manuscript entitled "Adsorption behavior and mechanism of the PFOS substitute OBS (sodium p-perfluorous nonenoxybenzene sulfonate) on activated carbon deals with removal of emerging environmental contaminant. The manuscript is written systematically. However, I could find some lacunae in the experiment. I recommend the manuscript for publication after addressing the following queries,

- 1) When you powder GAC to get UAC the specific surface area should increase as particle size is decreasing. But authors have mentioned that the specific surface area decreases which is contradictory to the existing knowledge. Authors should explain or recheck the surface area.
- 2) Why authors used KOH for R1-GAC and R2 GAC?
- 3) What are the plausible structure of R1-GAC and R2-GAC?
- 4) Give the EDX and FTIR data for R1-GAC and R2-GAC, to support the structure and justify higher adsorption rate .

Review form: Reviewer 3

Is the manuscript scientifically sound in its present form?

Yes

Are the interpretations and conclusions justified by the results?

Yes

Is the language acceptable?

Yes

Do you have any ethical concerns with this paper?

No

Have you any concerns about statistical analyses in this paper?

No

Recommendation?

Major revision is needed (please make suggestions in comments)

Comments to the Author(s)

In this paper, the adsorption behavior of OBS on activated carbon (AC) with different physical and chemical properties were investigated. Further investigation indicated that a larger pore size and smaller particle size can greatly enhance the adsorptive removal of OBS on AC in systems with co-existing other PFASs and organic matter. The regeneration method of AC were studied and summarized systematically. The acceptance of this work is therefore recommended after the major revision addressing the following comments.

1. Author should add the comparison table and make a comparison with some other adsorbents
2. The authors should highlight better in which sense their work is novel compared to previous literature.

3. The overall adsorption process may be jointly controlled by external mass transfer and intraparticle diffusion, hence, the adsorption kinetic data was further analyzed by Boyd model (ACS Appl. Mater. Interfaces, 2012, 4(11), 5749)
4. The adsorption equilibrium time up to 70h, Why?
5. From fig. 4, the Na⁺ and Ca²⁺ under different concentration almost have the same effect trend, why?

Decision letter (RSOS-191069.R0)

22-Jul-2019

Dear Professor Shi:

Title: Adsorption behavior and mechanism of the PFOS substitute OBS (sodium p-perfluorooxononylbenzene sulfonate) on activated carbon
Manuscript ID: RSOS-191069

The editor assigned to your manuscript has now received comments from reviewers. We would like you to revise your paper in accordance with the referee and Subject Editor suggestions which can be found below (not including confidential reports to the Editor). Please note this decision does not guarantee eventual acceptance.

Please submit your revised paper before 14-Aug-2019. Please note that the revision deadline will expire at 00.00am on this date. If we do not hear from you within this time then it will be assumed that the paper has been withdrawn. In exceptional circumstances, extensions may be possible if agreed with the Editorial Office in advance. We do not allow multiple rounds of revision so we urge you to make every effort to fully address all of the comments at this stage. If deemed necessary by the Editors, your manuscript will be sent back to one or more of the original reviewers for assessment. If the original reviewers are not available we may invite new reviewers.

Please also include the following statements alongside the other end statements. As we cannot publish your manuscript without these end statements included, if you feel that a given heading is not relevant to your paper, please nevertheless include the heading and explicitly state that it is not relevant to your work.

- Acknowledgements

RSC Associate Editor:

Comments to the Author:

Please also complete the request of Reviewer 2 as follows: Figure 3. Give 2 sigma error bars in the graph

RSC Subject Editor:

Comments to the Author:

(There are no comments.)

Reviewers' Comments to Author:

Reviewer: 1

Comments to the Author(s)

This article investigated the adsorption behavior and mechanism of sodium p-perfluorooxynonylbenzene sulfonate (OBS) on activated carbons with different physicochemical properties. The removal of OBS from water using activated carbons was attractive to readers. Adsorption kinetics, adsorption isotherms, effects of pH and ionic strength have been examined, and adsorbent regeneration and applicability in co-existing matter systems have also been investigated. The topic is relatively new, which is the first time to examine the adsorptive removal of OBS using activated carbons. This work is important for the practical engineering application of adsorption for removing emerging contaminants. I recommend for publication after minor revision according to the following comments.

1. The abstract could be improved by providing specific information regarding the quantitative data. For example, the adsorption capacity of R-AC? the regeneration percent of hot water and NaOH solution?
2. OBS was a typical alternative to PFOS, and its structural formula should be given, which is very important to know the structural difference compared with PFOS?

3. This paper evaluated the influence of co-existing PFASs on the adsorption of OBS on ACs. OBS in actual water environment always coexists with other PFASs. What is the selectivity of activated carbons for the target OBS?
4. Page 4, line 6, better to use 'Per- and polyfluoroalkyl substances (PFASs)', instead of 'Perfluoroalkyl and polyfluoroalkyl substances'.
5. Page 9, line 21-22, better to give a more detailed description of the analytical methods, e.g., blanks, the LOD of HPLC-UV, the preparation of mobile phase, the sample analysis time, etc.
6. Page 14, line 14, change "previous studies" to "a previous study".
7. Page 15, Lines 28-29 mention two methods previously tested for regeneration of spent AC used to treat PFOS. Better to consider citing additional research related to spent AC that was specifically used to treat PFAS.
8. Fig. 3(a), UAC, not BAC. Fig. 3(c), O1-GAC, not Q1-GAC.
9. The adsorption isotherms of OBS was reported, and the relative adsorption references should be cited, such as *Nanoscale*, 2016, 8, 15978-15987, *Inorganic Chemistry* 2019, 58, 11, 7255-7266, *Mater. Chem. Front.*, 2019,3, 224-232. *Dalton Trans.*, 2018,47, 2791-2798,

Reviewer: 2

Comments to the Author(s)

The manuscript entitled "Adsorption behavior and mechanism of the PFOS substitute OBS (sodium p-perfluorooctanesulfonate) on activated carbon deals with removal of emerging environmental contaminant. The manuscript is written systematically. However, I could find some lacunae in the experiment. I recommend the manuscript for publication after addressing the following queries,

- 1) When you powder GAC to get UAC the specific surface area should increase as particle size is decreasing. But authors have mentioned that the specific surface area decreases which is contradictory to the existing knowledge. Authors should explain or recheck the surface area.
- 2) Why authors used KOH for R1-GAC and R2 GAC?
- 3) What are the plausible structure of R1-GAC and R2-GAC?
- 4) Give the EDX and FTIR data for R1-GAC and R2-GAC, to support the structure and justify higher adsorption rate .

Reviewer: 3

Comments to the Author(s)

In this paper, the adsorption behavior of OBS on activated carbon (AC) with different physical and chemical properties were investigated. Further investigation indicated that a larger pore size and smaller particle size can greatly enhance the adsorptive removal of OBS on AC in systems with co-existing other PFASs and organic matter. The regeneration method of AC were studied and summarized systematically. The acceptance of this work is therefore recommended after the major revision addressing the following comments.

1. Author should add the comparison table and make a comparison with some other adsorbents
2. The authors should highlight better in which sense their work is novel compared to previous literature.
3. The overall adsorption process may be jointly controlled by external mass transfer and intra-particle diffusion, hence, the adsorption kinetic data was further analyzed by Boyd model(*ACS Appl. Mater. Interfaces*, 2012, 4(11), 5749)
4. The adsorption equilibrium time up to 70h, Why?
5. From fig. 4, the Na⁺ and Ca²⁺ under different concentration almost have the same effect trend, why?

Author's Response to Decision Letter for (RSOS-191069.R0)

See Appendix A.

RSOS-191069.R1 (Revision)

Review form: Reviewer 1

Is the manuscript scientifically sound in its present form?

Yes

Are the interpretations and conclusions justified by the results?

Yes

Is the language acceptable?

Yes

Do you have any ethical concerns with this paper?

Yes

Have you any concerns about statistical analyses in this paper?

No

Recommendation?

Accept as is

Comments to the Author(s)

This revised manuscript can be accepted in its current form.

Decision letter (RSOS-191069.R1)

14-Aug-2019

Dear Professor Shi:

Title: Adsorption behavior and mechanism of the PFOS substitute OBS (sodium p-perfluorous nonenoxybenzene sulfonate) on activated carbon

Manuscript ID: RSOS-191069.R1

It is a pleasure to accept your manuscript in its current form for publication in Royal Society Open Science. The chemistry content of Royal Society Open Science is published in collaboration with the Royal Society of Chemistry.

RSC Associate Editor:
Comments to the Author:
(There are no comments.)

RSC Subject Editor:
Comments to the Author:
(There are no comments.)

Reviewer(s)' Comments to Author:
Reviewer: 1

Comments to the Author(s)
This revised manuscript can be accepted in its current form.

Appendix A

Response to reviewer comments

(Manuscript ID: RSOS-191069)

The comments of the reviewers are very much appreciated and helped improve the manuscript significantly. We responded to all the comments and made all of the requested changes. Those changes are highlighted with red color in the revised manuscript. In the following section, we explained in details how we responded to each of the comments.

Response to RSC Associate Editor

Comments to the Author:

Please also complete the request of Reviewer 2 as follows: Figure 3. Give 2 sigma error bars in the graph

Response: Thanks for this good comment. We add 2 sigma error bars in Figure 3.

After revision:

Fig. 3.

Response to Reviewer #1

This article investigated the adsorption behavior and mechanism of sodium p-perfluorous nonenoxybenzene sulfonate (OBS) on activated carbons with different physicochemical properties. The removal of OBS from water using activated carbons was attractive to readers. Adsorption kinetics, adsorption isotherms, effects of pH and ionic strength have been examined, and adsorbent regeneration and applicability in co-existing matter systems have also been investigated. The topic is relatively new, which is the first time to examine the adsorptive removal of OBS using activated carbons. This work is important for the practical engineering application of

adsorption for removing emerging contaminants. I recommend for publication after minor revision according to the following comments.

1. The abstract could be improved by providing specific information regarding the quantitative data. For example, the adsorption capacity of R-AC? the regeneration percent of hot water and NaOH solution?

Response: Thanks for this comment. We add some specific information in the *abstract*.

After revision:

Page 2, Line 19: Reactivation of AC by KOH can greatly enlarge their pore size and surface area, greatly increasing their adsorption capacities. The adsorption capacity of two kinds of R-GAC exceeded 0.35 mmol/g, significantly higher than that of other ACs.

Page 2, Line 31: The spent AC can be successfully regenerated by methanol, and it can be partly regenerated by hot water and NaOH solution. The percentage of regeneration for the spent AC was 70.4% with 90 °C water temperature and up to 95 % when 5% NaOH was added into the regeneration solution.

2. OBS was a typical alternative to PFOS, and its structural formula should be given, which is very important to know the structural difference compared with PFOS?

Response: Thanks for this good comment. To make it clear, we added the important information in the section of *2.1 Chemicals and materials*.

After revision:

Page 5, Line 35: OBS was purchased from Wengjiang reagent Co. Ltd. (Guangzhou, China) and its structural formula was showed in Table S1†.

Table S1 Physicochemical properties of OBS

PFASs	Chemical formula	Molecular lengtha (nm)	Chemical structure
OBS	$C_9F_{17}OC_6H_4SO_3Na$	1.26	

3. This paper evaluated the influence of co-existing PFASs on the adsorption of OBS on ACs. OBS in actual water environment always coexists with other PFASs. What is the selectivity of activated carbons for the target OBS?

Response: Thanks for this comment. It is true that co-existing organic pollutants will significantly affect the sorption of target compounds on activated carbon. Actually, in this paper, we mainly focused on the study of the adsorption behavior and mechanism of OBS on activated carbon. The selectivity of adsorbents for the target OBS in the presence of other traditional pollutants in water will be further studied in our next study.

4. Page 4, line 6, better to use 'Per- and polyfluoroalkyl substances (PFASs)', instead of 'Perfluoroalkyl and polyfluoroalkyl substances'.

Response: Thanks for this good suggestion. We revised it.

5. Page 9, line 21-22, better to give a more detailed description of the analytical methods, e.g., blanks, the LOD of HPLC-UV, the preparation of mobile phase, the sample analysis time, etc.

Response: Thanks for this good suggestion. We add detailed description of the analytical methods in the section of *2.6 Analytical methods*

After revision:

Page 8, line 27: A methanol/0.02 M dihydrogen phosphate buffer solution (8: 2, v/v) was used as the mobile phase at a flow rate of 1.0 mL/min. The mobile phase was ultrasound for 30 min to fully blend and remove bubbles. Deionized water was used for blank determination. The LOD of HPLC-UV was 0.20 mg/L, and the analysis time of each sample was 8.5 min.

6. Page 14, line 14, change "previous studies" to " a previous study".

Response: Thanks for this good comment. We revised it.

7. Page 15, Lines 28-29 mention two methods previously tested for regeneration of spent AC used to treat PFOS. Better to consider citing additional research related to spent AC that was specifically used to treat PFAS.

Response: This is a good suggestion. Indeed, there are many regeneration methods for ACs. In

addition to organic solvents and hot water, salt solutions and advanced oxidation can also be used. We added some information in 3.5 *Adsorbent regeneration and applicability in co-existing systems*. But the most common method is organic solvents, the more environmentally friendly methods are salt solution and hot water, so we chose these three methods for experimentation in this study.

After revision:

Page 14, Line 43: ACs containing adsorbed PFOS can be regenerated using hot water, organic solvents, salt solutions and advanced oxidation [5, 20, 41].

References

[5] Deng S, Nie Y, Du Z, Huang Q, Meng P, Wang B, Huang J, Yu G. 2015 Enhanced adsorption of perfluorooctane sulfonate and perfluorooctanoate by bamboo-derived granular activated carbon. *J. Hazard. Mater.* 282, 150-157.

[20] Du Z, Deng S, Liu D, Yao X, Wang Y, Lu X, Wang B, Huang J, Wang Y, Xing B. 2016 Efficient adsorption of PFOS and F53B from chrome plating wastewater and their subsequent degradation in the regeneration process, *Chem. Eng. J.* 290, 405-413.

[41] Wang W, Du Z, Deng S, Vakili M, Ren L, Meng P, Maimaiti A, Wang B, Huang J, Wang Y, Yu G. 2018 Regeneration of PFOS loaded activated carbon by hot water and subsequent aeration enrichment of PFOS from eluent. *Carbon* 134, 199-206.

8. Fig. 3(a), UAC, not BAC. Fig. 3(c), O1-GAC, not Q1-GAC.

Response: Thanks for this good suggestion. We revised it.

9. The adsorption isotherms of OBS was reported, and the relative adsorption references should be cited, such as *Nanoscale*, 2016, 8, 15978-15987, *Inorganic Chemistry* 2019, 58, 11, 7255-7266, *Mater. Chem. Front.*, 2019,3, 224-232. *Dalton Trans.*, 2018,47, 2791-2798,

Response: Thanks for this good suggestion. We add the relative adsorption references in 3.3 *Adsorption isotherms*.

After revision:

Page 12, Line 5: The adsorption isotherms for OBS on the different ACs are shown in Fig. 3, together with the Langmuir and Freundlich models used to describe the isotherm data [28, 29].

Page 13, Line 9: The Langmuir equation assumes that during the adsorption process there is monolayer coverage of the adsorbates on the adsorbents with no interactions between the adsorbate molecules [31, 32].

References

[28] Zhang Y, Zhang M, Yang J, Ding L, Zheng J, Xu J, Xiong S. 2016 Formation of Fe₃O₄@SiO₂@C/Ni hybrids with enhanced catalytic activity and histidine-rich protein separation. *Nanoscale* 8(35), 15978-15988.

[29] Zheng J, Zhang M, Miao T, Yang J, Xu J, Alharbi NS, Wakeel M. 2019 Anchoring nickel nanoparticles on three-dimensionally macro-/mesoporous titanium dioxide with a carbon layer from polydopamine using polymethylmethacrylate microspheres as sacrificial templates. *Mater. Chem. Front.* 3(2), 224-232.

[31] He W, Guo X, Zheng J, Xu J, Hayat T, Alharbi NS, Zhang M. 2019 Structural Evolution and Compositional Modulation of ZIF-8-Derived Hybrids Comprised of Metallic Ni Nanoparticles and Silica as Interlayer. *Inorg. Chem.* 58(11), 7255-7266.

[32] Wang J, Zhang M, Xu J, Zheng J, Hayat T, Alharbi NS. 2018 Formation of Fe₃O₄@C/Ni microtubes for efficient catalysis and protein adsorption. *Dalton T.* 47(8), 2791-2798.

Response to Reviewer #2

The manuscript entitled "Adsorption behavior and mechanism of the PFOS substitute OBS (sodium *p*-perfluorous nonenoxybenzene sulfonate) on activated carbon deals with removal of emerging environmental contaminant. The manuscript is written systematically. However, I could find some lacunae in the experiment. I recommend the manuscript for publication after addressing the following queries.

1. When you powder GAC to get UAC the specific surface area should increase as particle size is decreasing. But authors have mentioned that the specific surface area decreases which is contradictory to the existing knowledge. Authors should explain or recheck the surface area.

Response: Thank you for this suggestion. We retested and checked the specific surface area results of UAC, and the results were consistent with the previous ones. Indeed, the specific surface area always increase with decreasing the particle size of adsorbent. However, mechanical crushing with too high strength will destruct the pore structure of activated carbon [23]. As shown in **Table 1**, the specific surface area of UAC is lower than that of GAC, because its pore structure has been

destructured during such high strength mechanical crushing process. To make the results more credible, we cited other references to support this statement.

After revision:

Page 9, Line 21: The specific surface area of GAC is little lower than that of PAC but higher than UAC (Table 1). The specific surface area of UAC is lower, because its pore structure has been destructed during such high strength mechanical crushing process [23].

References

[23] Meng P, Fang X, Maimaiti A, Yu G, Deng S. 2019 Efficient removal of perfluorinated compounds from water using a regenerable magnetic activated carbon. *Chemosphere* 224, 87-194.

2. Why authors used KOH for R1-GAC and R2 GAC?

Response: Thank you for this good comment. KOH has been found to be one of the most effective compounds to reactivate activated carbons [1, 2]. Precious studies also have used KOH, and the reactivated activated carbon showed enlarged pores and high surface area [3, 4].

References

[1] Lozano-Castello D, Lillo-Rodenas M, Cazorla-Amorós D, Linares-Solano A, 2001 Preparation of activated carbons from Spanish anthracite: I. Activation by KOH. *Carbon* 39, 741-749.

[2] Sudaryanto Y, Hartono S, Irawaty W, Hindarso H, Ismadji S, 2006 High surface area activated carbon prepared from cassava peel by chemical activation. *Bioresource technol.* 97, 734-739.

[3] Deng S, Nie Y, Du Z, Huang Q, Meng P, Wang B, Huang J, Yu G. 2015 Enhanced adsorption of perfluorooctane sulfonate and perfluorooctanoate by bamboo-derived granular activated carbon. *J. Hazard. Mater.* 282, 150-157.

[4] Wang W, Deng S, Li Z, Ren L, Shan D, Wang B, Huang J, Wang Y, Yu G. 2018 Sorption behavior and mechanism of organophosphate flame retardants on activated carbons. *Chem. Eng. J.* 332, 286-292.

3. What are the plausible structure of R1-GAC and R2-GAC?

Response: Thanks for this comment. Activated carbon is a kind of black powder, granular or pellet with porous amorphous carbon. ACs exhibit a heterogeneous pore structure [1]. Since AC is a disordered porous structure, we cannot give the definite structure of R-GAC, and we describe

the general structure of it. R-GAC was impregnated by KOH solution at KOH/C mass and heated at 900 °C under N₂ for 1.5 h. After treatment, the number of pores and the pore size of R-GAC greatly increased, shown in Fig. 1 and Fig. S2(b)†. In general, R-GAC also has a heterogeneous pore structure similar to activated carbon, but the pore size and the number of pores of R-GAC were increased after activation. To further explain the structure of R1-GAC and R2-GAC, we added some descriptions of R-GAC compared to GAC in this article.

After revision:

Page 9, Line 11: In general, R-GAC also has a heterogeneous pore structure similar to activated carbon, but the pore size and the number of pores of R-GAC were increased after activation. The enlarged pore sizes are favorable for the diffusion of molecular OBS into the granular adsorbent.

Reference

[1] Boehm H. 1994 Some aspects of the surface chemistry of carbon blacks and other carbons. Carbon 32, 759-769.

4. Give the EDX and FTIR data for R1-GAC and R2-GAC, to support the structure and justify higher adsorption rate.

Response: Thanks for this good suggestion. Because we do not have the instruments and conditions for EDX test, so we cannot give the EDX data for R-GAC. But we used an elemental analyzer to measure the elemental composition of the prepared ACs and the element composition and proportion of R1-GAC and R2-GAC are showed on **Table 1**. We also considered using FTIR to analyze the functional group on R-GAC and functional group changes before and after the adsorption of activated carbon (Fig. S3 and Fig.SA), to further illustrate its adsorption mechanism. However, no functional group changes were found on the surface of activated carbon before and after adsorption, so it's difficult to use FTIR to justify the high adsorption rate of R-GAC. The higher adsorption rates of R1-GAC and R2-GAC are due to their large pore volume and pore number. Still, to support the structure of R1-GAC and R2-GAC, the FTIR data was used to give a short description in the section of.3.1 *Characterization of prepared activated carbon*.

After revision:

Page 9, Line 43: Fig.S3† shows the FT-IR spectra of R1-GAC and R2-GAC. There are hydroxyl and carboxyl oxygen-containing functional groups on the surface of R-GAC.

Table 1

Adsorbent	Surface area	Pore volume	Elemental composition (%)				(O+N)/C (%)	Particle size (mesh)
	(m ² /g)	cc/g	C	H	N	O		
GAC	670.7	0.36	86.45	1.82	0.19	4.62	5.57	20-25
PAC	765.1	0.51	86.45	1.82	0.19	4.62	5.57	80-150
UAC	632.5	0.43	85.74	2.01	0.17	7.79	9.29	Fig. S1
R1-GAC	1108.6	0.62	91.63	1.19	0.13	3.45	3.91	20-25
R2-GAC	1705.1	0.89	88.41	1.35	0.03	5.42	6.17	20-25
O1-GAC	727.9	0.44	81.64	2.35	0.78	7.95	10.70	20-25
O2-GAC	725.5	0.44	78.78	2.32	0.90	10.81	14.87	20-25

Fig. S3 FTIR spectra of R1-GAC and R2-GAC

Fig. SA (a) FTIR spectra of R2-GAC before and after OBS (b) FTIR spectra of R1-GAC before and after OBS adsorption

Response to Reviewer #3

Comments to the Author(s)

In this paper, the adsorption behavior of OBS on activated carbon (AC) with different physical and chemical properties were investigated. Further investigation indicated that a larger pore size and smaller particle size can greatly enhance the adsorptive removal of OBS on AC in systems with co-existing other PFASs and organic matter. The regeneration method of AC were studied and summarized systematically. The acceptance of this work is therefore recommended after the major revision addressing the following comments.

1. Author should add the comparison table and make a comparison with some other adsorbents

Response: Thanks for this suggestion. Because OBS is an emerging alternative to PFOS, its adsorption removal on other adsorbents has not been investigated. Our study is the first one to explore the adsorption removal of OBS on activated carbon, and there is no other literature on OBS adsorption removal for reference. We are so sorry that we cannot make a comparison with some other kinds of adsorbents. This is also a major innovation in our research. We chose the activated carbon adsorbent to study the adsorption removal of OBS because of it's the most commonly used in the water treatment of tap water.

2. The authors should highlight better in which sense their work is novel compared to previous literature.

Response: Thank you for this good suggestion. OBS has been widely used as a substitute of PFOS, which has been banned, and has the same serious harm to human health, biology and ecological environment. However, as far as we know, there are only two papers discussing OBS removal from water using an oxidation method and aeration-foam collection. Our study is the first one to investigate the adsorption removal of OBS on activated carbon. And we add some information in *1. Introduction* to highlight the innovation points of this study.

After revision:

Page 5, Line 21: Our study is the first one to investigate the adsorption behavior and mechanism of OBS on activated carbon.

3. The overall adsorption process may be jointly controlled by external mass transfer and

intra-particle diffusion, hence, the adsorption kinetic data was further analyzed by Boyd model (ACS Appl. Mater. Interfaces, 2012, 4(11), 5749)

Response: Thank you for this good suggestion. To further study the adsorption mechanism of OBS on ACs, we used Boyd model to analyze the adsorption kinetic data and add discussion in the section of 3.2 *Adsorption kinetics*.

After revision:

Page 11, Line 37: The Boyd model has been widely used for studying the mechanism of adsorption [27], and Fig.S5† showed the Boyd plots for the first 24 h adsorption of OBS on ACs. The plots showed a nonlinear segment before sorption equilibrium, suggesting that the rate of adsorption was not only controlled by pore diffusion in the initial period and chemical reaction also controlled the rate of adsorption.

Fig. S5

Reference

[27] Ma J, Yu F, Zhou L, Jin L, Yang M, Luan J, Tang Y, Fan H, Yuan Z, Chen J. 2012 Enhanced adsorptive removal of methyl orange and methylene blue from aqueous solution by alkali-activated multiwalled carbon nanotubes. ACS Appl. Mater. Inter. 4(11), 5749-5760.

4. The adsorption equilibrium time up to 70h, Why?

Response: Thank you for this suggestion. Our previous research has found that ACs with a bigger particle size have longer pores and less exposed sites for OBS adsorption, resulting in a correspondingly slower adsorption process [1]. Due to the big size of GAC and the big molecular size of OBS, it took long time for OBS to diffuse into the long pores of GAC due to the steric

hindrance effect in diffusion process. And we add some discussion in 3.2 *Adsorption kinetics*.

After revision:

Page 9, Line 11: The longer equilibration time for GAC was similar to our previously reported data for the adsorption of organophosphate flame retardants on activated carbon [24]. Due to the big size of GAC and the big molecular size of OBS, it took long time for OBS to diffuse into the long pores of GAC. The ACs with a larger particle size have deeper pores and less exposed sites for OBS adsorption, resulting in a correspondingly slower adsorption process.

Reference

[1] Wang W, Deng S, Li D, Ren L, Shan D, Wang B, Huang J, Wang Y, Yu G. 2018 Sorption behavior and mechanism of organophosphate flame retardants on activated carbons. *Chem. Eng. J.* 332, 286-292.

5. From fig. 4, the Na⁺ and Ca²⁺ under different concentration almost have the same effect trend, why?

Response: Thank you for this suggestion. We checked the data and redid the experiment, and sorry for incorrect result of ionic strength experiment. All the data except the ionic strength experiment were measured by HPLC and verified to be correct. In our previous ionic strength experiment, we used the UV spectrophotometer to test the results because it is easy to block the column of HPLC by considering the high concentration of salt ions in the sample. Because the salting-out effect was very serious, the OBS concentration was very low at high salt concentration, and the UV spectrophotometer test results were not accurate, result in giving the wrong experimental results. After inspection and re-experiment, we retested the samples using HPLC-UV and revised in 3.4 *Effect of solution pH and ionic strength* and **Fig. 4(b)**. Because the OBS concentration after adsorption was around the LOD when the Ca²⁺ concentration was 100 mmol/L, we deleted the results of 100 mmol/L Ca²⁺ and Na⁺.

After revision:

Page 14, Line 17: The removal of OBS by PAC increased when increasing the Na⁺ concentration from 0 to 50 mmol/L, and the higher increase in the removal of OBS in the presence of Ca²⁺ could be attributed to the stronger salting-out effect of Ca²⁺ when compared to Na⁺ [36]. This is consistent with previous work [37] which showed that Ca²⁺ showed a stronger influence than Na⁺ on the adsorption of PFASs on PAC. Moreover, the solubility of OBS decreased sharply at high

salt concentration, behavior which is similar to PFOS [38]. The aggregation of PAC can be formed due to the squeezing-out effect derived from ionic strength [39], and electrostatic screening can also cause the weak electrostatic repulsion among PAC particles and aggregation of PAC [40]. These two opposing effects caused by the solutions ionic strength were possibly involved in the OBS adsorption on ACs. However, since the salt-out effect exhibited a more significant effect than PAC aggregation, OBS removal rate increased with the increase of ionic strength.

Fig. 4.